# Facilitating Community Transition to Sustainable Land Governance: A Study of a Communal Settlement in South Africa

Nicholas Pinfold and Masilonyane Mokhele *



Department of Urban and Regional Planning, Faculty of Informatics and Design, Cape Peninsula University of Technology, Cape Town 8000, South Africa; pinfoldn@cput.ac.za
* Correspondence: mokhelem@cput.ac.za; Tel.: +27-21-440-2246

**Abstract:** Land is a fundamental resource that provides a foundation for the economy. Despite a wide range of studies on land governance systems, there is a lack of literature that analyzes the ability of communities to manage a change to different land governance systems. The study aimed to analyze the potential for the Goedverwacht communal settlement in the Western Cape province, South Africa, to transition from a hierarchical governance structure to one based on a communal land governance system. This aim was addressed by answering the research question: What are the roles, expectations and management strategies of the institutions and stakeholders participating in land governance? The study considered the community's desire to maintain its communal settlement's existence, and the choice between communal or individual freehold land governance. To understand these issues, the study utilized a framework that includes three theories: the theory of planned behaviour, the theory of institutional capacity, and the critical theory. (2) Methods: Through a survey, qualitative interviews, and focus group discussions, the study analyzed various underlying factors that influenced land governance and the land governance system desired by the community. (3) Results: The findings reveal that power dynamics and conflicting interests significantly affected the community's ability to manage potential modernization resulting from land reform. While establishing land rights can positively impact economic growth and social mobilization, the lack of the communal settlement's central government's capacity to manage modernization effectively can lead to instability. (4) Conclusions: The paper concludes that balancing institutionalization and modernization is crucial for effectively managing the transition to new land governance systems.

**Keywords:** communal settlement; land governance; land reform; community ownership; sustainable development

## 1. Introduction

Land is a fundamental resource that provides a foundation for the economy and the well-being of societies [1,2]. Through Sustainable Development Goal indicator 1.4, the United Nations has set a target to "ensure that all men and women, in particular, the poor and the vulnerable, have equal rights to economic resources, as well as access to basic services, ownership, and control over land and other forms of property . . . " [3]. The land reform program in South Africa comprises three elements of redistribution, restitution, and land tenure, which are considered critical land acquisition and allocation mechanisms. Rural land tenure involves conflicting land and property usage, development, transfer, and inheritance practices. Two conflicting viewpoints characterize the public discourse on rural land tenure and tenure security. One perspective advocates for total land tenure security within the communal land system. The other perspective holds that titling (i.e., the issuance of title deeds) is necessary to guarantee tenure security. Land reform typically involves legal and policy changes to recognize and secure communities' land rights. These changes may involve transferring land ownership from private landowners to the community

through different legal mechanisms. Once the landowner consents to the establishment of a settlement on their land, the community can establish a communal property association, and the owner's level of involvement would hinge on the agreement's specifics [4]. As the word community is ambiguous, it should be noted that the paper adopts a simplified interpretation of community as a group of people living in the same area.

When people have secure land rights, they are more likely to invest in that land, make improvements, and use it as collateral for loans and mortgages [1,2]. This commitment can increase productivity and economic growth. Furthermore, people with secure land rights are more likely to participate in decision-making processes related to the management and use of the land. This participation can increase civic engagement and empowerment and contribute to more equitable and sustainable development. However, if political stability and accountability are lacking, it can be difficult for people to exercise their rights and engage in civic activities [5,6].

Land reform is a pressing issue in many countries, and the process of implementing it can sometimes occur rapidly without adequate preparation or support for the communities affected. This unpreparedness can lead to challenges for the communities in adapting to and managing change effectively [7]. Numerous studies have been conducted to analyze land reform and reconcile formal and informal land governance systems for enhancing land tenure security in communal settlements [2,8–18]. These studies provide an overview of the key themes in land tenure and property rights in the global South. They highlight the tension between the dominant private property paradigm and informal land governance systems that provide tenure security but lack official recognition.

Despite a wide range of studies on land reform and land governance systems, there is a lack of literature that analyzes the ability of communities to manage a change to different land governance systems. The study therefore aimed to assess the potential for a communal settlement to transition from a hierarchical governance structure to one based on communal land ownership and decision-making, and to suggest a development path to ensure a sustainable transition to community land ownership and decision-making. As discussed in Section 3, the study focused on the communal settlement of Goedverwacht in South Africa. Goedverwacht is a Moravian Church mission station, which accommodates approximately 500 households.

This study's aim was addressed by answering the research question: What are the roles, expectations and management strategies of the stakeholders participating in land governance? As detailed in Section 2, the paper utilizes a conceptual framework that considers a range of factors, which include the beliefs of individuals and groups towards a specific type of landholding, the capacity of the institutions and stakeholders involved, and the economic, social, and political factors that shape land governance. The paper also highlights the need to balance institutionalization and modernization to achieve a sustainable shift from a hierarchical governance structure.

The paper is structured as follows: Section 2 reviews the literature, including establishing the conceptual framework that informed the study. Section 3 outlines the research methods employed to answer the research question. Section 4 presents the study's findings, followed by a discussion of how the community could transition to a system of communal land ownership and decision-making. Section 6 concludes the paper.

## 2. Literature Review

This section presents the three crucial elements constituting the conceptual framework for evaluating the community's determination to shift from a hierarchical governance structure to a communal land ownership and decision-making system. These elements are the community's belief in a specific land governance system, the community's institutional capacity to adjust to a different land governance system, and the structures that influence land governance and development within a community. The interconnection of these elements serves as the starting point for establishing a conceptual framework to guide the analysis.

## 2.1. Belief

If the residents of a communal settlement decide they want to live a life of communalism rather than individualism, it signifies their allegiance to the culture of their community. Communalism involves a hybrid land governance system in which land allocation, disputes, and use are controlled by the community within the overall land governance system of the state. A positive attitude and subjective norm towards communalism and communal land ownership are critical factors in determining the success of this type of land holding. A community that takes responsibility for its basic services will enhance the sustainability of the communal settlement. In contrast, when there is conflict among the residents regarding land holding, it can hinder effective governance and create significant obstacles.

In a study evaluating cadastral systems in periods of uncertainty, Barry [19] suggests using the theory of planned behaviour to assess the individuals' or households' intention to conform to the official cadastral system. Barry [19] considered the theory of reasoned action, but found it limited by the requirement of volitional control over behaviour, which is too narrow for studying the use of the cadastral system. Instead, the theory of planned behaviour, which includes attitude, subjective norm, and perceived behavioural control as determinants of behavioural intention, was adopted. The theory of planned behaviour addresses incomplete volitional control by considering the effects of perceived behavioural control, which is the third antecedent of intention in the theory of reasoned action. Volatile control is an essential aspect of this study in asserting residents' attitudes and subjective norms towards communal land ownership and decision-making. The individual's intention is important, but behaviour can only be expressed under volitional control. Research informed by the theory of planned behaviour often employs quantitative methods, such as surveys, to measure individuals' attitudes, subjective norms, and perceived behavioural control regarding a specific behaviour or outcome. However, it is common for these studies to utilize qualitative methods, such as interviews or focus groups, to gain a more in-depth understanding of the causes of the behaviour and the context in which it occurs [10,11].

## 2.2. Institutional Capacity

Various studies [2,12,13,15] express the importance of institutional capacity as a critical factor in securing land rights. The theory of institutional capacity is relevant in this study to assess the capacity of formal and informal institutions to manage the transition successfully.

Hornby, Royston, Kingwill, and Cousins [15] have researched institutional capacity. Their studies were initiated in response to the failure of numerous communal property institutions created in South Africa after 1994, highlighting the need to evaluate tenure security in the context of communal property associations and land-holding trusts. Their research methodology was participatory and interactive, and aimed at analyzing the nature of the problems regarding land tenure in rural villages. One of the key outcomes of the research was the realization of the complexity and dynamism of how people access land and claim rights to it, highlighting the need to build institutions around land rights that include customary rights. The position of the academics involved in the Learning Approach to Securing Tenure (LEAP) project was that formal land tenure with title deeds does not necessarily provide tenure security to communal settlements. Instead, if well organized, informal and customary systems can provide tenure security. Therefore, the LEAP project focused on policy interventions to formalize tenure arrangements in the case studies by issuing title deeds, but the research found that locally embedded property relations could not easily be transformed into registered tenures. Thus, the LEAP project highlighted the importance of building institutional capacity to support land tenure security in rural areas.

Beinart, Delius, and Hay [2] focused on enhanced governance and stewardship and the sustainability of land tenure, titling, and restitution issues in South Africa. The academics argue against traditionalist and non-democratic sources of authority, and they believe this is polarizing the forward-looking initiatives. It is argued that land ownership and administration under traditionalist intermediaries do not accommodate change. Their work relates to institutional capacity in the context of land governance. The focus is on

building the capacity of the state to develop land administration institutions that can support customary tenure and protect land rights. Their research identifies a lack of state capacity at the national level as a problem in adopting hybrid systems of land governance. It is suggested that capacitating the state to develop land administration institutions could bridge the gap between customary and statutory law, and enhance the governance and stewardship of land tenure, titling, and restitution issues. The proposed solutions, such as fit-for-purpose land administration systems, require the state to have adequate institutional capacity to implement and maintain them effectively.

Barry's [9] investigation centred on institutional capacity by gauging the efficacy of the South African cadastral system in fulfilling the community's needs during periods of change. The study examined the correlation between tenure and the cadastral system, evaluating the formulation of land policies and land administration. It also delved into the theories of planned behaviour and social change models to evaluate the system's effectiveness during social, political, and economic transformation. The analysis indicated that while the cadastral system is appropriate, acknowledged, and utilized during stable conditions, it may not be fully employed during volatile situations. This highlights the necessity of enhancing the institutional capacity to guarantee the system's effectiveness during such circumstances.

The institutional capacity theory looks at how institutions can help or hinder development. It implies that institutions have a significant role in creating a society's political, social, and economic outcomes [1,20,21]. The institutional capacity theory differs from the modernization theory in emphasizing the role of institutions, notably the state, in fostering or obstructing growth. It contends that a strong state is required for societal growth and that modernization should begin by amassing state power [22]. This is a different perspective from modernization theory. The issues expressed by Huntington [5] regarding the consequences of modernization on political institutions and the need to examine their relationships to understand the progress or deterioration of political institutions can be applied to the community, which is changing. Similarly, Fukuyama [23] and Huntington [24] expressed relevant matters about the dominance of individual rights and values and the conflict that arises from culture and religion. The institutional capacity theory provides insight into these political processes.

Fukuyama [23] and Huntington [24], both political analysts, offer further perspectives on the socio-political progress of the community and its impact on land administration. Their concepts aid in clarifying the obstacles that the community confronts in balancing personal and collective identities, as well as cultural and religious principles. These tensions may have implications for institutions that support land tenure security, as different groups may have varying views on how land should be owned, used, and governed. According to Fukuyama, the diversity of civilizations in the world has concluded and succeeded by novel ones. In Fukuyama's [23] view, the concept of individual rights has become the dominant organizing principle of modern societies. Identities formed around ethnicity, race, or religion have reduced in importance, allowing individuals to select their own identity. Fukuyama [23], therefore, claims that the era of ideological struggle has ended. The challenge of society today is managing the tensions between the diversity of individual identities and interests. Huntington [24], on the other hand, argues that the differences between the values and beliefs of cultures and religions would become the primary source of conflict. Huntington [24] emphasizes culture and religious identity as sources of conflict, whereas Fukuyama [23] emphasizes individual identity and the dominance of liberal democratic values.

### 2.3. Structure

The critical realist framework is a philosophical perspective that aims to provide explanations not just for what is occurring, but also for why and how it is occurring. In so doing, it probes the potential risks and benefits of the phenomenon being studied. In the context of this study, the power dynamics and conflicts may impact the community transition [25].

Critical realism provides a valuable lens to analyze the mechanisms behind successful or unsuccessful organizational change. Sayer [26] criticizes the attempts of critical social science and critical realist philosophy to derive normative conclusions based on critiques of social phenomena. Sayer argues that they do not adequately address the difficulties in justifying critical standpoints and finding alternative social forms that generate fewer problems than the ones they replace. They should not be used to derive normative conclusions (what ought to be the case) because they lack a robust framework for doing so. However, Sayer [26] does not entirely reject their usefulness and acknowledges that critical social science and critical realism provide a valuable critique of existing social structures and beliefs. The studies conducted by Beinart, Delius, and Hay [2] touch on the issue of institutional capacity as a critical factor in securing land rights and understanding the underlying social, economic, and political dynamics that shape land tenure systems. According to Hornby, Kingwill, Royston, and Cousins [15], this involves understanding power relations, social structures, and cultural norms.

Causal mechanisms produce or trigger observable events [27]. Mechanisms unrelated to an event are trans-empirical, explaining why observable occurrences occur [28]. The causative mechanisms cannot be directly determined because they cannot be observed. Empirical research and theory development can help us understand causal mechanisms. The basic idea behind critical realism is to build deeper layers of explanation rather than identifying general laws, as is the case in positivism [29]. The goal is to uncover the underlying mechanisms that have produced or could produce occurrences of interest. In the actual world, events occur due to the interaction of social structures, mechanisms, and human agency [29]. Causal mechanisms can alter situations; nevertheless, the mechanism's realization is governed by the conditions under which it operates. Mechanisms should be viewed as tendencies induced by underlying causal mechanisms rather than empirical generalizations [30].

A critical realist framework was employed in the study to examine the underlying causes and mechanisms. These mechanisms can potentially impact situations, but their actual influence is determined by the varying conditions in which they operate [31]. These mechanisms should be viewed in terms of the tendencies they generate rather than as empirical generalizations [30]. Causal mechanisms are abstract concepts that explain the underlying reasons and processes that lead to specific events or outcomes. They cannot be observed directly but are inferred through empirical research.

*2.4. A Synthesis: Towards Conceptual Framework for Communal Land Governance and Development*

There are several factors to consider when determining a conceptual framework for analyzing a communal land governance system. The land and the community are at the heart of communal governance; these two elements are mutually reinforcing. The land offers the community a location to live, work, and obtain food. In turn, the community members manage and maintain the land, assuring its production and protecting its ecological and cultural worth. Appropriate governance structures and procedures may improve collaborative decision-making, encourage accountability, and assure equitable benefit and cost distribution [2,15].

Conversely, inadequate governance can lead to conflict, depletion of resources, and the unequal distribution of benefits in communal land development. This process involves intricate interactions among economic, social, and environmental factors. Alterations in land tenure systems, for instance, can influence access to resources and power relations within the community, which may result in positive or negative consequences. Additionally, communal land management and advancement outcomes vary and depend on specific contextual factors, including cultural norms and values, resource availability, political and economic circumstances, and historical and geographical contexts.

Accordingly, a standardized approach is unlikely to yield satisfactory results, and resolutions should be customized to each community's unique requirements and desires.

Feedback loops between these factors and ideas are crucial to comprehend communal land management and progress. For example, adjusting governance frameworks can prompt land use and administration procedure modifications, which can, in turn, impact land productivity and community welfare. Similarly, shifts in the economic landscape can influence the community's governance structures and development preferences.

A framework that overlooks the distinct traits and circumstances of a specific setting may not be valuable or relevant. Therefore, it is imperative to meticulously assess the presumptions made in any framework and contemplate alternative theoretical frameworks that may be more pertinent. One possible premise in this framework for communal land governance and development is that the community bears the primary responsibility for communal land governance and development. Although community involvement is vital for successful communal land governance and development, other stakeholders may be involved, such as local governments, NGOs, and external stakeholders. Depending on the specific context under investigation, these entities may be crucial in shaping communal land governance.

Another supposition is that effective governance structures and processes will inevitably result in favourable outcomes. While effective governance is significant, the connection between governance and development outcomes may be more intricate. Other factors, such as resource availability, cultural norms and values, and power dynamics, may influence the results. Alternative frameworks that prioritize these other factors may be more applicable in specific settings. Furthermore, the assumptions made in the framework may be less applicable in settings where communal land governance and development are disputed or where multiple stakeholders have conflicting interests. In such cases, frameworks that consider power dynamics, conflict resolution mechanisms, and negotiation strategies may be more fitting.

The three components of the conceptual framework for communal land governance and development are the belief in communal land governance, the institutional ability to manage it correctly, and the understanding of the undercurrents that influence community growth. These components are crucial for executing long-term community land reform that lays the foundation for economic growth, social mobilization, and accountability. Additional factors such as political will, social dynamics, and economic concerns can also influence communal land governance and development efficacy. The framework is under the umbrella of the relevant theories presented in Sections 2.1–2.3 namely the theory of planned behaviour, the theory of institutional capacity, and the critical theory (Figure 1).

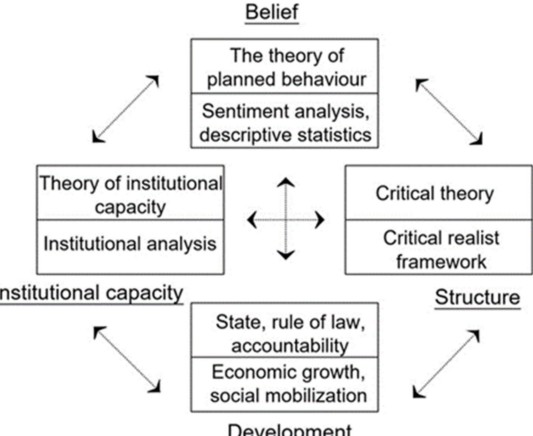

**Figure 1.** The conceptual framework for evaluating the community's determination to shift from a hierarchical governance structure to a communal land ownership and decision-making system.

There are intertwined relationships between the framework's components; for example, the belief in communal land governance may result in establishing social norms that promote collaboration, trust, and collective action. This, in turn, can help to foster social

cohesiveness, which is necessary for efficient community land governance. Furthermore, institutions must be capable of managing community land governance to guarantee that the appropriate legal and regulatory underpinnings for successful administration are in place. This helps to ensure that processes for monitoring and enforcing compliance, promoting openness and accountability, and resolving disputes are in place. For example, a lack of institutional capacity might erode the trust in community land administration, and ignoring the undercurrents that affect trust within the community can inhibit development. As a result, the three components concurrently accomplish successful community land reform.

The conceptual framework is built on three underlying assumptions. The first assumption posits that a person's attitude, subjective norms, and perceived control may impact their motivation to participate in community land governance, a crucial component of successful communal land reform. The second assumption is that the effectiveness of governance can be evaluated by examining the institutionalization of the political system. The strength and effectiveness of institutional structures, regulations, and procedures can significantly impact the success of communal land reform. The third assumption is that understanding the root causes of social, economic, and political issues can aid in identifying strategies to address them and promote positive outcomes.

The scope of the conceptual framework is focused on a communal settlement's ability to shift from a hierarchical land governance system to a communal land governance system. This implies that the framework is intended to handle the transformation's unique difficulties and possibilities. According to the conceptual framework, a community will be able to successfully transition from a hierarchical governance structure to a communal land governance system if it believes in a communal way of life and has the governance capacity to manage communal land governance, as well as the absence of negative causal structures and mechanisms that impede development. Additionally, such communities can successfully manage the modernization that comes with land reform. Political deterioration may occur if the community lacks the institutional ability to handle modernization, such as economic expansion and social mobilization. The benefits of modernization may be lost, resulting in turmoil and perhaps impeding the community's capacity to accomplish long-term sustainability.

The conceptual framework is intended to provide a structured and systematic approach to assessing the suitability of a communal settlement to become a communal property association. The framework can identify areas that may require support or resources to ensure the success of communal property associations, particularly the community's capacity to manage and govern the communal property effectively, including the availability of technical expertise, leadership, and institutional capacity. Additionally, the framework can be used to assess the level of social cohesion and inclusivity in the community by examining whether the community shares the same vision and values and the extent of trust and cooperation among community members.

There are several restrictions to the framework that necessitates consideration. Firstly, the framework is founded on presumptions that may not universally apply to different settings and communities. For instance, the assumption that people's intentions to live a particular lifestyle are determined by their attitudes, subjective norms, and perceived behavioural control may not be entirely accurate, and other factors, such as social and economic inequality, historical and cultural factors, and power dynamics, may also play a role. Secondly, the framework may not wholly encompass the intricacies and diversity of communities and their governance and development requirements. Communities are not uniform and may have varying priorities, values, and interests, necessitating different governance and development approaches. Thirdly, the framework may not account for the broader political and economic landscape in which communities are positioned and how these surroundings may shape and confine governance and development possibilities. To counter these limitations, it is vital to tailor the framework to specific contextual settings and continuously reflect and revise as new information and perspectives arise. This could involve conducting detailed research, consulting with communities to ascertain their needs,

priorities, and challenges, and improving the framework accordingly. It could also involve ongoing conversation and collaboration with other stakeholders, including government agencies, civil society organizations, and private sector actors, to establish consensus and support for governance and development initiatives.

## 3. Methods

### 3.1. Study Area

The study focused on the communal settlement of Goedverwacht in the Western Cape province of South Africa, within the Bergriver municipality (Figure 2).

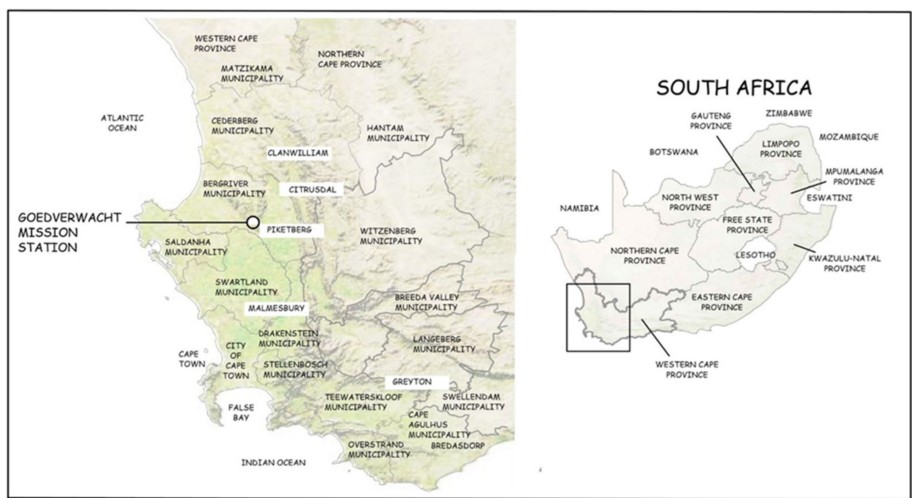

**Figure 2.** Location of the study area.

The mission station offers housing for Moravian Church members and smallholder farms for lease from the church. Accommodating approximately 500 households, the village is not subjected to formal planning regulations and private property rights and is organically growing in a linear pattern along the Platkloof River (Figure 3).

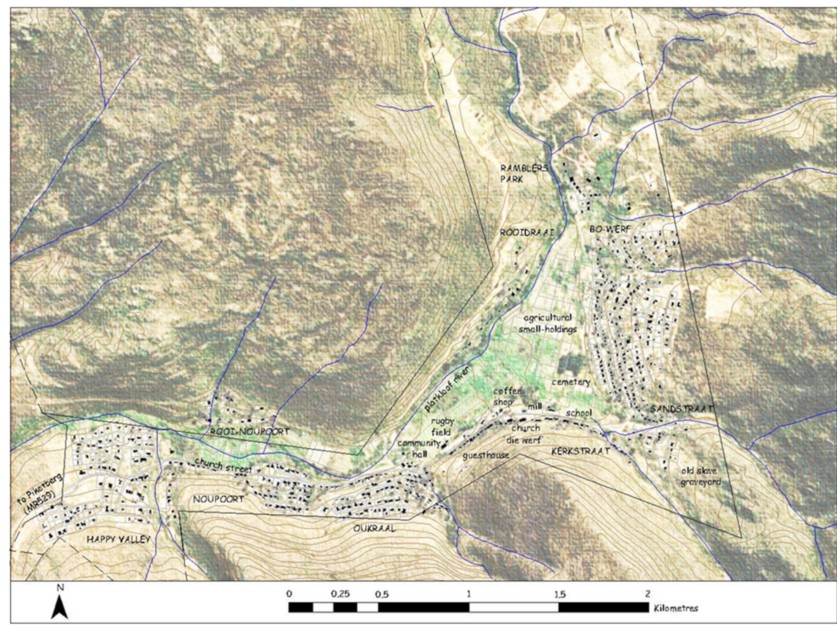

**Figure 3.** Layout of the Goedverwacht mission station.

Goedverwacht is governed by a community-elected Overseers Council, accountable to the local Moravian Church. Properties on the settlement are free but with monthly fees

for essential services, infrastructure, and municipal rates. However, the actual monthly fee paid by the community could not be established. The church leadership has expressed that they are considerate towards collecting fees from the indigent, but the issue lies in the community's accustomed payment of a fixed amount, regardless of their income level. To address this, the church plans to implement a sliding scale system considering the individual's income.

The Goedverwacht community was selected for this research as it is contemplating transforming from a hierarchical governance structure into a self-governing community, with residents having more authority over the land as part of a land reform program.

### 3.2. Data Collection and Analysis Methods

A mixed-method approach, comprising quantitative and qualitative aspects, was used in the study. A framework that includes three theories was used to guide the data collection and analysis. These are the theory of planned behaviour, the theory of institutional capacity, and the critical theory discussed in Section 2. The theory of planned behaviour provides the framework for understanding the community's belief towards communal living. This helps to identify the factors that may influence the willingness of the community to persist with a communal lifestyle. The theory of institutional capacity is relevant to assess institutions' capacity to manage the transition successfully. Formal and informal institutions are considered part of a land governance system. Critical theory is pertinent to understanding the underlying structures and mechanisms that influence community development's economic, social, and political aspects.

Data Collection Methods

The study employed a combination of a survey, qualitative interviews, and focus group discussions on gaining insights into land rights and governance in Goedverwacht. These data collection methods are presented below.

Household Survey

A survey was conducted to gather insights into the respondents' attitudes, perceptions, and behaviours related to communal living and land governance in the study area. The questionnaire was divided into two parts. The first part consisted of open-ended questions about the individual's experiences of living in the village. Several socio-political variables of interest, including satisfaction with the governance of the community, were noted. These variables were used to guide the analysis later in the study. The second part of the questionnaire comprised a Likert rating scale to measure the respondents' attitudes, opinions, and perceptions towards communal living, in line with the theory of planned behaviour [32].

Sixty-nine heads of household were interviewed of the approximately 500 households (14%). All participants were 21 years or older, with 55% below the age of 65 and 45% over the age of 65 and retired. The interviews were conducted face-to-face on 24 and 25 February 2022. Simple random sampling was used to select a sample of households from the mission station. Each household was assigned a unique identifier using Geographic Information System (GIS) software. Once each household had a unique identifier, the GIS software generated a random sequence of numbers corresponding to the household identifiers using the software's random number generator function. After the random sequence of numbers was generated, the corresponding households were selected from the dataset.

Assuming a confidence level of 95%, a margin of error of 5%, and a population size of 500 households, a sample size of approximately 385 households was envisaged. The researchers achieved an equal spread across the nine neighbourhoods in the village. However, the response rate was low due to the COVID-19 pandemic restrictions, with most heads of household unavailable during the survey. It also became evident that the community had a high degree of similarity or sameness in terms of characteristics, implying a saturation point had been reached. The saturation point occurs when the researchers feel they have collected enough data to answer the research question, and the effort required

to obtain new information is disproportionate to the insights gained [33]. Despite the saturation point being reached, it is acknowledged that the response rate achieved makes the study susceptible to non-response bias [34].

Focus Group Discussions

Focus group discussions were conducted in line with the methodology outlined by Hennink [35]. Two focus groups were conducted, the first with three representatives from various groups within the community and the second with six representatives from various groups within the community and a facilitator. The community groups included the Goedverwacht Komitee GGK, the Khoi Cochoqua Clan Elders, and the 'Klower' owned Goedverwacht Awakens NPC.

Each focus group discussion lasted approximately two hours. Three participants from the second focus group discussion also participated in the first. A discussion guide was developed and translated into Afrikaans, and the focus group discussions were conducted in both English and Afrikaans, the languages used in the community. The discussions included a series of open-ended questions to elicit the participants' experiences and opinions on the governance of the village. These were designed to allow participants to express their views and to obtain various responses. The discussions were intended to activate detail that may have been overlooked using other data collection methods. Group discussions overcome self-consciousness and encourage participants to disclose information. The discussions were a debate on land governance that presented differing and conflicting views. The discussion was mainly amongst the participants, displaying interesting group dynamics. The participants were able to build on each other's comments, which provided an in-depth view of the situation. Some unforeseen remarks and criticisms added value to the research.

Qualitative Interviews

In addition to the survey and focus group discussions, qualitative interviews were conducted with:

- Community leaders, namely the reverend of the local Moravian Church, a member of the overseer's council (governing body), and representatives of the Goedverwacht Komitee GGK, the Khoi Cochoqua Clan Elders, and the 'Klower' owned Goedverwacht Awakens NPC.
- Municipal manager of the Berg River municipality.
- A spatial planner from the Berg River municipality.

The face-to-face interviews, conducted in November 2021, were guided by a semi-structured questionnaire entailing open-ended questions (Table 1).

**Table 1.** Interview guide.

| | |
|---|---|
| 1 | What is the community's attitude towards land governance? |
| 2 | Does social pressure influence the community's decisions regarding land governance |
| 3 | To what extent does the community feel land governance is controlling? |
| 4 | Is the 'Opsienesrad' (supervisory committee) able to manage land issues adequately? |
| 5 | Has the 'Opsienesrad' (supervisory committee) been able to adapt to modern circumstances? |
| 6 | Does the 'Opsienesrad' (supervisory committee) have an appropriate organizational structure? |
| 7 | Is there unity within the 'Opsienesrad' (supervisory committee)? |
| 8 | Does the 'Opsienesrad' (supervisory committee) support collaborative governance? |
| 9 | What are the community norms and values regarding land? |

**Table 1.** *Cont.*

| 10 | What stakeholders are central to local land governance? |
|----|---------------------------------------------------------|
| 11 | Is there a political agenda behind off-register transactions? |
| 12 | Is there an informal land market in the community? |
| 13 | What is your attitude towards local land governance? |
| 14 | Does social pressure influence your decisions regarding land governance? |
| 15 | To what extent do you feel local land governance is controlling? |

*3.3. Analytical Methods*

The analysis of the intention of the community to live a communal life involved combining both quantitative and qualitative methods. The survey was analyzed using descriptive statistics such as mean, standard deviation, and confidence intervals for each statement or inquiry to determine the level of agreement or disagreement with each statement. In addition, correlation coefficients were calculated between attitude, subjective norm, perceived behavioural control, and the resident's intention to carry out the behaviour to measure the strength of the relationship between these variables. The purpose of this enquiry was threefold; firstly, it assisted the researchers in understanding the degree of influence each variable had on the resident's intention to live a communal life. Secondly, it helped to identify the strength and direction of the relationships between variables. An increase in one variable may lead to an increase in the other. Finally, it helps establish the reliability and validity of the survey instrument.

The data from the qualitative interviews were analyzed using content analysis. This involved coding the transcribed text of the interviews into attitude, subjective norm, and perceived behavioural control. Once the transcribed text was coded, sentiment analysis was conducted to determine each code's negative, neutral, and positive sentiments. The results were intersected to establish a more comprehensive understanding of the relationships between the variables studied.

Institutional analysis was conducted using Woodhill's institutional analysis tool to investigate the roles, power dynamics, and impact of the two predominant institutions governing the community: the Moravian Church and the community association. The four concepts of meaning, control, action, and association were investigated using open-ended questions to understand the institutional capacity of the community to implement land reform. The degree of institutionalization within the political institution of the Moravian Church was then conducted using the four criteria outlined by Huntington [5]: flexibility, complexity, autonomy, and coherence. This method is intended to determine the rise or decline of institutionalization within a society under a particular political system by defining and measuring the benchmarks. The goal was to analyze the community's institutional capability and strike a balance between institutionalization and modernity, which is suggested before land reform.

The researchers used a critical realist framework to examine the underlying causal structures and mechanisms that impede progress in land reform.

## 4. Findings

*4.1. Attitude of the Residents*

The first part of the investigation focused on the residents' intended behaviour towards the type of lifestyle they wanted to lead. The theory of planned behaviour was used to guide the data collection methods (survey and interviews) and analysis. The survey was used to measure the residents' attitudes, subjective norms, and perceived behavioural control. The survey included eight statements or inquiries that were succeeded by a set of alternative responses on a Likert scale of 7 points, which indicates the level of agreement or disagreement with the statement (see Tables 2 and 3, Figure 4).

**Table 2.** Statements or inquiries in the 'Head of Household' survey.

| A1 (instrumental attitude) | Your way of life provides you with security. |
|---|---|
| A2 (experiential attitude) | You are passionate about your 'way of life'. |
| A3 (affective attitude) | You enjoy participating in community activities. |
| SN1 (descriptive norm) | Most people think and feel the same way as you do. |
| SN2 (injunctive norm) | Others in the community support your view. |
| SN3 (injunctive norm) | Your family and friends support your view. |
| PBC (capacity and autonomy) | You can achieve your goals. |
| NT (intention) | You support a communal lifestyle. |

**Table 3.** Results of the 'Head of Household' survey (LS1 disagree, LS4 neutral, LS7 agree).

| S | N | LS1 | LS2 | LS3 | LS4 | LS5 | LS6 | LS7 | Mean | SD | CL | SCI |
|---|---|---|---|---|---|---|---|---|---|---|---|---|
| A1 | 69 | 6 | 1 | 0 | 1 | 6 | 5 | 50 | 5.07 | 11.55 | 95% | 4.77 to 5.37 |
| A2 | 69 | 4 | 3 | 3 | 7 | 14 | 11 | 27 | 4.22 | 9.46 | 95% | 3.98 to 4.46 |
| A3 | 69 | 2 | 0 | 2 | 6 | 5 | 4 | 50 | 4.72 | 12.39 | 95% | 4.38 to 5.06 |
| SN1 | 69 | 25 | 5 | 0 | 3 | 3 | 2 | 31 | 3.36 | 9.92 | 95% | 3.10 to 3.62 |
| SN2 | 69 | 6 | 4 | 3 | 10 | 9 | 8 | 29 | 4.13 | 9.15 | 95% | 3.90 to 4.36 |
| SN3 | 69 | 4 | 2 | 0 | 1 | 6 | 4 | 52 | 5.29 | 13.43 | 95% | 4.90 to 5.68 |
| PBC | 69 | 31 | 2 | 1 | 5 | 8 | 5 | 17 | 2.83 | 8.55 | 95% | 2.62 to 3.04 |
| INT | 69 | 2 | 0 | 0 | 3 | 4 | 5 | 55 | 5.58 | 13.13 | 95% | 5.17 to 5.99 |

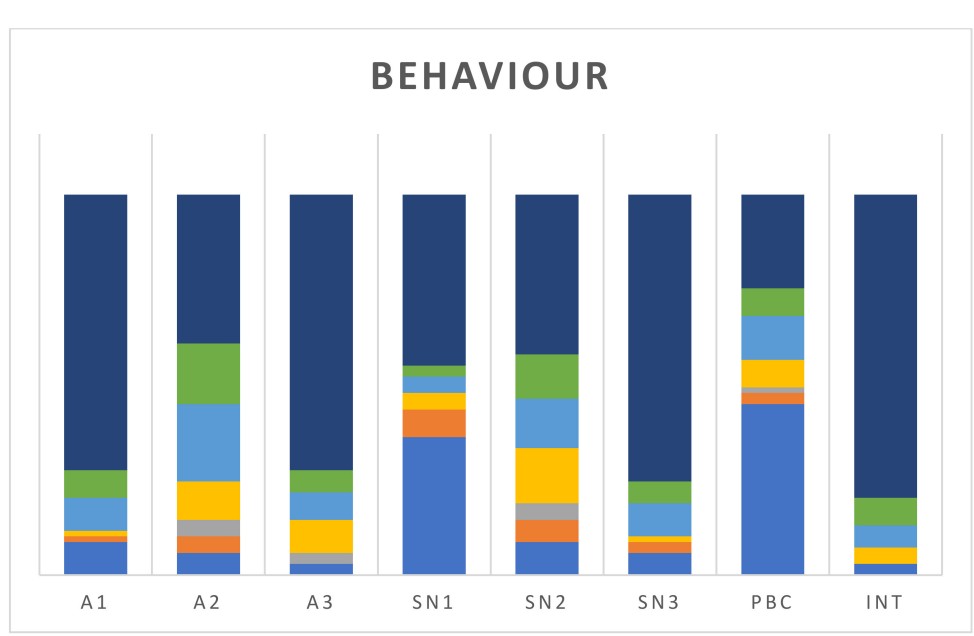

**Figure 4.** Results of the 'Head of Household' survey.

Analyzing the correlation between attitudes and intentions towards communal land governance and economic, social, and political factors allows a more holistic understanding of the community's perspective. This approach unveils the intricate interplay between individual attitudes, social norms, and broader contextual elements, offering valuable insights into the community's overall stance on communal land governance. These correlations show how various attitudes and norms are interconnected with the community's intention to participate in communal land governance. By considering these correlations alongside

economic, social, and political considerations, a more comprehensive comprehension of the community's disposition towards land governance can be attained (Table 4).

**Table 4.** Correlation coefficients between attitude/subjective norm and intention.

| |
|---|
| (A1) Instrumental attitude—(INT) Intention to carry out the behaviour r = 0.9946 |
| (A2) Experiential attitude—(INT) Intention to carry out the behaviour r = 0.9106 |
| (A3) Affective attitude—(INT) Intention to carry out the behaviour r = 0.9973 |
| (SN1) Descriptive norm—(INT) Intention to carry out the behaviour r = 0.7314 |
| (SN2) Injunctive norm—(INT) Intention to carry out the behaviour r = 0.9760 |
| (SN3) Injunctive norm—(INT) Intention to carry out the behaviour r = 0.9965 |
| (PBC) Perceived behavioural control (capacity and autonomy)—(INT) Intention to carry out the behaviour r = 0.3025 |

The qualitative data collected through the survey included an open-ended question asked to the 'head of household' participants: How likely are you to continue living a communal lifestyle in the next year considering your personal beliefs, the opinions of people important to you, and the control you have over your circumstances? The transcribed interviews were coded into four codes in Atlas.ti [36]. The text was assigned the codes of attitude, subjective norm, and perceived behavioural control. There was some overlap in the coding, such as attitude and subjective norm (19%), attitude and perceived behavioural control (6%), and subjective norm and perceived behavioural control (6%). This overlap indicates that they are related and may influence each other. Sentiment analysis was then conducted to establish whether each category was negative, neutral, or positive (see Table 5). The results revealed that the respondents were positive about living a communal lifestyle; however, their perceived behavioural control was negative. The intersection of the qualitative results supports the quantitative results.

**Table 5.** Sentiment analysis of the 'Head of Household' survey.

| Code | Negative | Neutral | Positive | Total |
|---|---|---|---|---|
| Attitude | 6 | 6 | 25 | 37 |
| Subjective norm | 1 | 6 | 9 | 16 |
| Perceived behavioural control | 9 | 4 | 3 | 16 |

### 4.2. Institutional Capacity

Analyzing the relationship between institutions is necessary to understand their impact on human behaviour. To effect positive societal changes, it is essential to comprehend institutional innovation. It is important to acknowledge that in addition to institutional analysis, power dynamics and conflicts in a community can also be conveyed and resolved through non-institutional methods, such as attitudes, cultural norms, personal relationships, and informal networks. To analyze non-institutional means, the researchers employed the household survey. Furthermore, a critical realist framework was utilized to analyze the causal structures and mechanisms involved.

The Moravian Church and the community association were two of the predominant institutions in the community, the Moravian Church with its governing council and the community association comprising various groups in the community. Woodhill's [37] institutional analysis was used to understand the two institutions' roles, power dynamics, and impact on the community (Figure 5). Woodhill's [37] institutional analysis tool comprises four key concepts: meaning, control, association, and action.

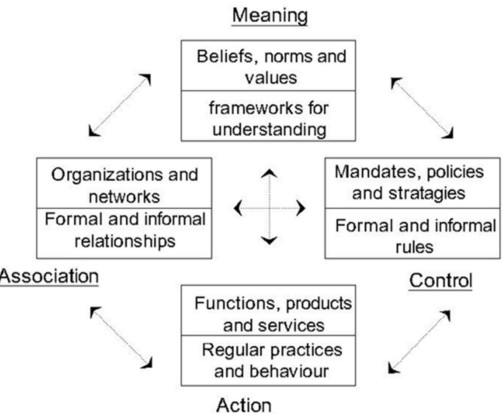

**Figure 5.** A framework for exploring the complexity of institutions [36].

The four concepts of meaning, control, action, and association were investigated to understand the institutional capacity of the community to implement land reform. The following open-ended questions were used to analyze the institutions. The first question concerned: 'What beliefs, norms and values shape the institutions?' The framework of meaning was used as a lens to understand how institutions and the community interpreted their circumstances. The second question was about control: 'How do formal and informal institutions contribute to the control of the community?' The concept of control refers to rules that govern behaviour. The third question was to do with action: 'What practices and behaviour of the institution contribute to achieving its overall goals and objectives?' This question was used to understand how the institutions carried out their work and how their activities contributed to their overall goals and objectives. The fourth question was to do with the association: 'What networks within the institution influence decision-making?' Networks can play a significant role in shaping the policies and practices of individual institutions. The results are shown in Table 6.

**Table 6.** Results of the institutional analysis.

| Institution Domain | Church Overseers Council (Formal) | Community Association (Informal) |
|---|---|---|
| Meaning—beliefs, norms, and values (a framework for understanding) | Christian values<br>Ideological outlook<br>Seek political autonomy<br>External influence (rotation of clergyman) | Seek collaborative non-corrupt governance<br>Utopian outlook<br>Indigenous/cultural affiliation<br>Represents the community |
| Control—mandates, policies, and strategies (formal and informal rules) | Enforce the rules and control conflict authority (ignore the Community Association)<br>Collect taxes | Agitate for the land rights<br>Economic development plan<br>Oppose the Overseers Council (legal action)<br>Seek to resolve conflict through dialogue |
| Association—organisations and networks (relationships and transactions) | The local Moravian Church<br>Local government<br>Family structures—heads of household | Community groups—social and moral issues<br>Tourism |
| Action—functions, products, and services (regular practices and behaviour) | Governance<br>Basic service delivery<br>Infrastructure development<br>Control of resources | Promote tourism and festivals<br>Job creation<br>Social mobilisation |

Huntington [5] evaluates the degree of institutionalization using four criteria: flexibility, complexity, autonomy, and coherence. By defining and measuring these benchmarks, it is feasible to determine the rise or decline of institutionalization within a society under a certain political system. This is important in analyzing the community's institutional capability and balancing institutionalization and modernity.

The recorded responses were transcribed and analyzed using the Atlas.ti software. The text was classified and organized into four categories. Sentiment analysis was performed to determine whether the community's institutions met, partially met, or did not meet the requirements (see Table 7).

**Table 7.** Final assessment of institutionalisation.

| Criteria | Open-Ended Question | Complies/Partially Complies/Does Not Comply |
|---|---|---|
| Adaptability | Can the community adapt to a changing environment or a new challenge? | Partially complies |
| Complexity | Has the community's organization changed over time? | Partially complies |
| Autonomy | Is there a lack of consensus or unity among community members regarding the community's functional limits and conflict resolution procedures? | Partially complies |
| Coherence | Do community members lack consensus or unity regarding the community's functional limits and conflict resolution procedures? | Partially complies |

Based on the findings, the community institution only partially met each of the criteria reviewed. These findings indicate that the institutions suffered from some issues, but also had strengths and potential areas for progress.

*4.3. Economic, Social, and Political Factors That Shape Land Governance*

A thorough investigation into the community's economic, social, and political conditions was deemed necessary to understand the obstacles hindering land reform comprehensively. A critical realist framework was used to delve deeper into the underlying causal structures and mechanisms impeding progress in land reform. While the previous section on institutional capacity addressed issues such as political stability and the institutions supporting land reform, further attention was needed to address the power dynamics and the interests of different stakeholders. Additionally, the economic issue of employment had not received sufficient attention. To address these issues, focus group discussions and interviews were conducted to enquire about employment opportunities and the power dynamics and interests of the stakeholders. An open-ended question was posed during the focus group discussions and interviews to prompt the participants to share their experiences and observations regarding the impact of power dynamics and stakeholder interests on the economic situation in the community: 'Can you share any experiences or observations on how power dynamics and stakeholder interests have affected the economic situation in this community?'.

The transcribed text from the focus group discussions with community members was coded as economic, political, or other. Themes were then established from the coded text (see Table 8).

**Table 8.** Summary of content analysis of focus group discussions with community groups.

| Code | Theme | Sentiment |
|---|---|---|
| Economic | Potential for self-sustainability through agriculture | Positive |
| | Need for community vision and self-governance reliance | Positive |
| | Concerns about Church's involvement in community projects | Negative |
| | Lack of transparency and community input in decision-making | Negative |
| Political | Community development | Neutral |
| | Land Ownership and land reform | Neutral |
| | Role of Institutions in addressing community needs | Neutral |
| | Lack of trust in the Church to support community entrepreneurial activities | Negative |

The transcribed text from the interview with the church leadership was coded as economic, political, or other. Themes were established from the coded text (see Table 9).

**Table 9.** Summary of content analysis of interviews with the church leadership.

| Code | Theme | Sentiment |
|------|-------|-----------|
| Economic | Succession of knowledge | Neutral |
| | Water reticulation maintenance | Neutral |
| | Skills development | Positive |
| Political | Church and community governance | Neutral |
| | Agricultural management | Neutral |
| | Committee and community service | Neutral |
| | Land Ownership and land reform | Neutral |
| | Self-governance and community empowerment | Positive |

The analysis of the focus group discussion indicated a positive outlook for achieving self-sustainability through agriculture, as well as a community vision of self-governance and self-reliance. However, there were negative sentiments towards the church's involvement in community projects and a lack of support for community entrepreneurial activities. Despite the contentious nature of land reform, the sentiment surrounding it was neutral, possibly due to the support for land reform from the church. The sentiments associated with other issues, such as the role of the institutions in addressing community needs, were mixed, encompassing both positive and negative perceptions. The interviews with the church leadership yielded constructive feedback, and the overall sentiment expressed was neutral regarding the situation in the community. There was a positive response to upskilling the population to improve service delivery. The maintenance of the water reticulation system was a concern, as was the transfer of knowledge regarding the maintenance of services within the community; however, there was confidence that these issues could be resolved. The governance of the community, agricultural management, and the selection of the overseer's council were issues discussed, with the sentiment surrounding it being neutral. The church was open to land reform in the community; however, there were differences in opinion regarding the form that the land reform should take. The church positively supported self-governance and community empowerment. The church's positive sentiment towards self-governance and community empowerment and the community's positive sentiments towards a community vision and self-governance reliance align with the results of the behavioural survey, which support a communal lifestyle. The results were as follows. Empirical domain: (1) observing community meetings revealed new initiatives being proposed and discussed—the previous survey results revealed that the community's attitude was positive towards their type of lifestyle; (2) complaints or grievances filed against institutions that are perceived to be unresponsive to community needs—the previous survey results revealed that the community's perceived behavioural control was low; (3) instances of verbal confrontations, disagreements, and conflicts that arose due to the perceived lack of support for community entrepreneurial activities by the church. Actual domain: (1) increased participation in decision-making processes; (2) apathy and disengagement from community members who did not feel that their voices were heard or that the institutions were addressing their needs; (3) This leads to increased tensions and conflict between the church and the community, as well as a lack of collaboration and cooperation towards shared goals. Real domain: (1) community empowerment (positive); (2) community governance and institutional role (mixed sentiment); (3) church's involvement in community projects and the lack of support for community entrepreneurial activities (negative) (see Figure 6).

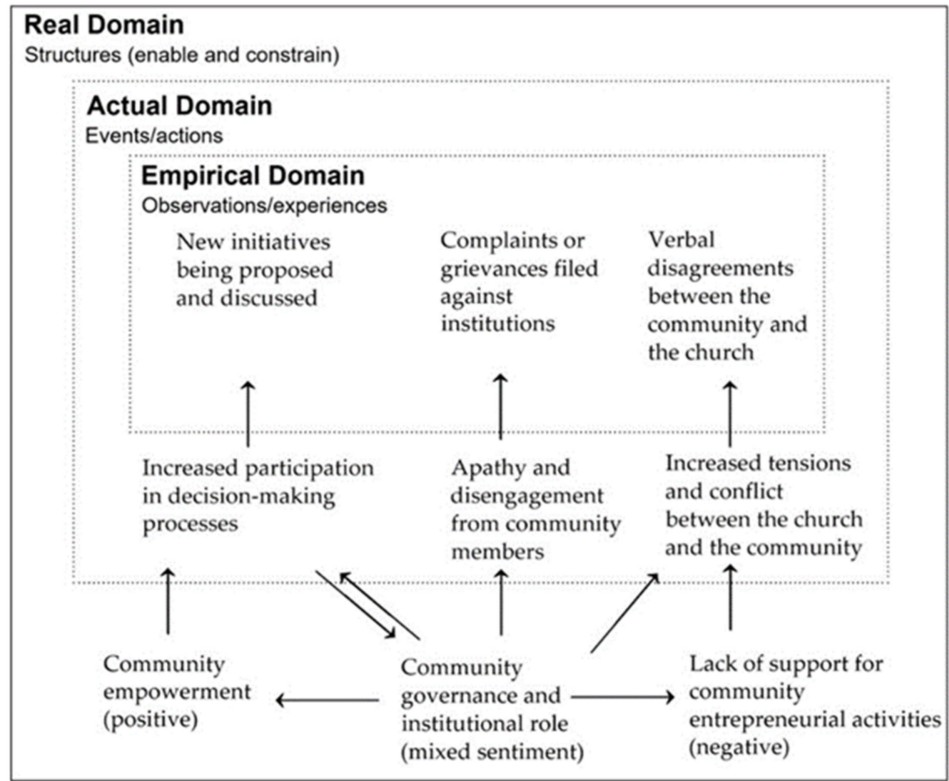

**Figure 6.** Critical realist framework analysis. Adapted from Anderson [31].

*4.4. Summary of the Findings*

Regarding the institutional analysis, the church and the community association were the two predominant institutions involved in land governance in the community. To answer the question of what roles, expectations, and management strategies the stakeholders involved in land governance have, the following has been deduced: The church operates formally, with a governing council, while the community association is informal and comprises various community groups. Formal and informal institutions contributed to the control of the community, with the church enforcing rules and controlling conflict, and the community association agitating for land rights and opposing the church council. The study also explored the beliefs, norms, and values that shaped these institutions, revealing that, on one hand, the church operated on Christian values and an ideological outlook, seeking political autonomy and responding to external influences, such as the rotation of clergymen. The community association, on the other hand, sought collaborative governance, had a utopian outlook and indigenous/cultural affiliation, and represented the community.

The study analyzed the institutional capacity of the church and the community association in implementing land reform, focusing on four essential components: adaptability, consistency, compatibility, and coherence. Based on Huntington's evaluation criteria, both institutions partially met the institutionalization standards. Hence, there is a need to improve their institutional capacity to implement land governance policies effectively. As actors involved in land governance in the community, the church is a formal institution that enforces rules and manages conflicts, while the community association is an informal institution that advocates for land rights and conflict resolution through dialogue. Both institutions must enhance their capacity in various areas to ensure successful land reform.

The community's stakeholders participating in land governance included community members and the church. Community members played a crucial role in decision-making processes and have a positive outlook towards self-sustainability through agriculture and a vision of self-governance and self-reliance. The church also played an essential role in the community, supporting self-governance and community empowerment, although negative sentiments existed regarding their involvement in community projects and their

lack of support for community entrepreneurial activities. Institutions also played a role in addressing community needs, although the sentiment surrounding their role was mixed, encompassing positive and negative perceptions.

The anticipated roles of the stakeholders involved in land governance encompassed heightened involvement in decision-making processes, community empowerment, and addressing grievances filed against unresponsive institutions. Management strategies should emphasize community empowerment and self-governance, as both the community members and the church expressed these positive sentiments. The church should strive to support community entrepreneurial activities and address reservations about their engagement in community projects. Meanwhile, institutions should tackle the mixed sentiment surrounding their function in addressing community needs and endeavour to foster greater collaboration and cooperation towards common objectives.

The economic, social, and political factors that shape land governance include the economic impact of transitioning to a common property resource management system. The social factors include the community's desire to maintain its communal settlement's existence and the choice between communal land governance or individual freehold land governance, as well as power dynamics and conflicting interests among community members. The political factors include the role of the church's central government in managing modernization and the potential for chaos and instability if this is not conducted effectively. The study highlights the need to balance institutionalization and modernization to manage the transition to new land governance systems effectively.

## 5. Discussion

This study introduces a conceptual framework that depicts how belief in communalism, institutional capacity, and underlying causal structures affect community development. By scrutinizing these aspects, the framework underscores the need to balance institutionalization and modernization to achieve a sustainable shift from a hierarchical governance structure to communal land ownership and decision-making system.

Based on the findings, ideology plays a crucial role in the Moravian Church's pursuit of its principles and values. It could be argued that the church has a conservative governance approach and does not allow sufficient participation from community members. Conversely, the residents hold a more hopeful outlook for the future. The research employed Francis Fukuyama's developmental framework to map the community's developmental trajectory, encompassing three developmental dimensions: economic growth, social mobilization, and legitimacy. Political development is included in this framework, with the state, rule of law, and accountability being key components. Fukuyama's development framework provides a good understanding of community development and can be crucial in contextualizing the transition to a new land governance system.

The development framework outlines the interconnected aspects of community growth through six dimensions. The first dimension underlines the state, and the second dimension highlights the state's role in facilitating economic growth, which fosters social mobilization in the third dimension. As the middle class expands, the third dimension asserts their tendency to demand accountability and democracy. In turn, the fourth dimension represents legitimacy, which is an essential component of the political order because it allows people to accept the decisions made by the government or governing institutions as valid and binding. Ultimately, the fifth dimension reinforces that the rule of law checks the state's power [6] (see Figure 7).

In terms of the five dimensions of development, the results suggest the following development map: Firstly, to encourage economic growth through a strong coherent state, the institutional capacity of both the church and the community association needs to be improved to implement land governance policies effectively. This can be achieved by enhancing their adaptability, consistency, compatibility, and coherence, as recommended by Huntington's evaluation criteria.

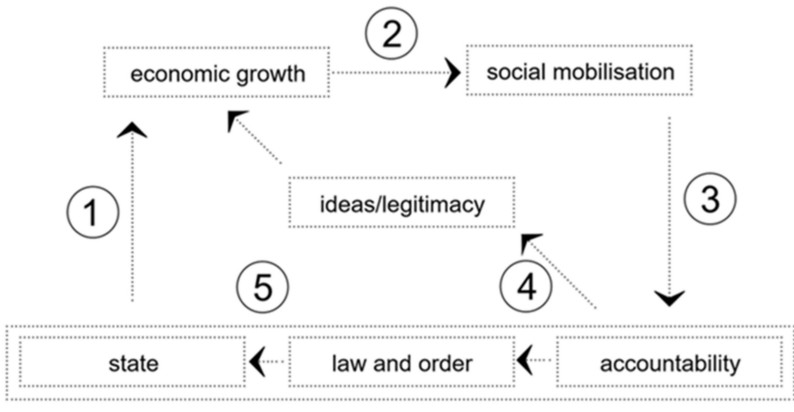

**Figure 7.** Five dimensions of the development framework: 1—a strong coherent state encourages economic growth; 2—the state resides over economic growth that results in social mobilization; 3—the larger middle class demands accountability and democracy; 4—democracy creates legitimacy and strengthens the rule of law; 5—the rule of law controls the state. Adapted from Fukuyama [6].

Secondly, to ensure that the state resides in economic growth that results in social mobilization, decision-making processes should involve community members as they play a crucial role in land governance. Empowering and self-reliant communities lead to economic growth, and the church should support community entrepreneurial activities.

Thirdly, to respond to the greater middle-class demands for accountability and democracy, decision-making processes need to involve a broader range of stakeholders, including middle-class representatives.

Fourthly, democracy creates legitimacy and strengthens the rule of law. Effective communication and engagement with stakeholders, including community members, leaders, and other relevant parties, are essential to achieve ideas and legitimacy. Building trust and credibility through transparency and accountability is also crucial for gaining legitimacy.

Lastly, transparency and accountability in land governance need to be increased to ensure that the rule of law controls the state. Mechanisms for community members to monitor and report on the performance of institutions involved in land governance can be created. Additionally, constructive dialogue with the church could address negative sentiments surrounding their involvement in community projects and their lack of support for community entrepreneurial activities.

Francis Fukuyama's development framework is criticized as being oversimplified, as it reduces the complex and multifaceted nature of a community's development trajectory into three dimensions; namely, economic growth, social mobilization, and legitimacy. The linear relationship assumed between dimensions may also be seen as overly simplistic, as the causal relationships between these dimensions can be much more complex and dynamic. Additionally, it is felt that the framework does not consider other important factors that can affect a community's development trajectory, such as culture, history, and geography. However, the framework provides a broad understanding of the development of a community, which can help understand the context in which a transition in land governance takes place. Tailoring the framework to a community's specific circumstances would enhance its usefulness, as can its focus on community engagement and inclusiveness in decision-making.

Fukuyama's development framework is more fitting when a community moves from a hierarchical governance structure to communal land ownership and decision-making. This is because the community may lack a clear understanding of their economic and political systems, and a structured and systematic approach to development is needed. It also helps to create a shared understanding among various stakeholders when resources for a culturally sensitive approach are limited. On the other hand, a culturally sensitive approach is more appropriate when local knowledge, values, and traditions are crucial to a

development project's success. An alternative economic approach, such as sustainable development, is suitable where balancing economic growth with environmental sustainability and social justice is essential. Ultimately, the approach should depend on the community's unique circumstances and needs.

## 6. Conclusions

The paper's objective was to evaluate the capacity of a communal settlement to transition from a hierarchical governance structure towards a system of communal land ownership and decision-making. To achieve this objective, the authors devised a conceptual framework that considered the community's belief in communal lifestyle, their institutional capability to govern land communally, and the causal structures and mechanisms that impact the development of the community positively or negatively.

The key finding of this study was that several factors play a crucial role in determining the ability of a community to shift from a hierarchical land governance system. These factors include a supportive outlook towards communalism, a shared subjective norm, a sense of control over the future, and the institutional capacity to manage the transition, while comprehending the underlying currents that impact development.

The securing of land rights has a positive impact on both economic growth and social mobilization. However, if the central government in a communal settlement cannot effectively manage modernization, it can lead to disorder and instability. To reap the benefits of modernization, the community must have established institutions that can provide a stable and efficient governance structure.

This study gives insight into the issue of land reform and its impact on communal settlements transitioning to communal property associations with shared land ownership and decision-making. It highlights the need for caution in choosing a land governance system that is manageable and not overwhelming. This research emphasizes the significance of institutional capability and the necessary infrastructure in supporting and safeguarding land governance decisions. However, the study's scope is limited to only one case study, and it is necessary to validate the conceptual framework in various contexts.

**Author Contributions:** Conceptualization, N.P. and M.M.; methodology, N.P. and M.M.; software, N.P.; validation, N.P. and M.M.; formal analysis, N.P.; investigation, N.P.; writing—original draft preparation, N.P. and M.M.; writing—review and editing, M.M. and N.P.; visualization, N.P.; supervision, M.M.; project administration, N.P. All authors have read and agreed to the published version of the manuscript.

**Funding:** This research received no external funding.

**Data Availability Statement:** Not applicable.

**Conflicts of Interest:** The authors declare no conflict of interest.

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
