# Peer review of "Facilitating Community Transition to Sustainable Land Governance: A Study of a Communal Settlement in South Africa"

_land, doi:10.3390/land12061132_

Round 1

Reviewer 1 Report

Important and interesting research work about land management local community-based. The manuscript describes the relationship between landowner governance over smallholder farmers' performance and life quality. the methodology used is the appropriate one, but some aspects of this should be improved to show the real significance and representation mean of the results. The lack of this information produces “weak” findings and discussion which should be improved.

Qualitative interviews: Please, show the questions done (insert in the text or as additional information).

Focus group: More information is needed about their specific composition and the role of each component to a better understanding of the results and their interpretation.

Line 1199. How much are the fees? This data could be important to analyze the community's economic benefit. As the authors point out, economic growth is important.

Line 221. “Lukit” scale? Or the Likert scale. Please check it.

Line 223.  Only 14% of 5000 households were interviewed. Is this number enough representative to of the whole community? Standard error and statistical probability (5%, 10%) to the confidence level of sample significative representation should be calculated and shown. For a good data/results interpretation, it is necessary to prove the meaning of the sample done concerning the whole community. Following this, another question is: Do all households have the same socioeconomic conditions? The 14% studied represent a balanced sample of these differences/socioeconomic conditions?

Figure 3. If you used a Linkert scale, please show the correspondence with agree/disagree level in Figure 3. I supposed that the scale is 1-7 but should better show the correspondence to a better understanding of answer analysis and findings obtained.

Line 442.  The authors talk about the unbalanced (unequal distribution) of benefits in communal land development. Please show some figures about this fact, they can improve the research findings and conclusions.

Author Response

Qualitative interviews: Please, show the questions done (insert in the text or as additional information).

The questions for the qualitative interviews have now been included in Section 3.2.1.3 (Table 1) and also highlighted in the findings section.

Focus group: More information is needed about their specific composition and the role of each component to a better understanding of the results and their interpretation.

The details of the community groups have been added to Section 3.2.1.2. The community groups included the Goedverwacht Komitee GGK, the Khoi Cochoqua Clan Elders, and the ’Klower’ owned Goedverwacht Awakens NPC.

Line 1199. How much are the fees? This data could be important to analyze the community's economic benefit. As the authors point out, economic growth is important.

The actual fee paid by the community members could not be established. Even the church did not appear transparent and could not provide information relating to the fees. The view of the church on the fees has however been included in Section 3.1.

Line 221. “Lukit” scale? Or the Likert scale. Please check it.

‘Lukit’ scale has been corrected to ‘Likert’ scale throughout the manuscript.

Line 223.  Only 14% of 5000 households were interviewed. Is this number enough representative to of the whole community? Standard error and statistical probability (5%, 10%) to the confidence level of sample significative representation should be calculated and shown. For a good data/results interpretation, it is necessary to prove the meaning of the sample done concerning the whole community. Following this, another question is: Do all households have the same socioeconomic conditions? The 14% studied represent a balanced sample of these differences/socioeconomic conditions?

In Section 3.2.1.1, it has been clarified that assuming a confidence level of 95%, a margin of error of 5%, and a population size of 500 households, a sample size of approximately 217 households was envisaged.  

It has now been acknowledged in Section 3.2.1.1 that, due to a low response rate (largely due to Covid-19-related reasons), the study is prone to a non-response error. As the community showed high similarity, Section 3.2.1.1 also argues that the saturation point might have been reached during the survey.

Figure 3. If you used a Linkert scale, please show the correspondence with agree/disagree level in Figure 3. I supposed that the scale is 1-7 but should better show the correspondence to a better understanding of answer analysis and findings obtained.

Has been added in Section 4, Table 3

Line 442.  The authors talk about the unbalanced (unequal distribution) of benefits in communal land development. Please show some figures about this fact, they can improve the research findings and conclusions.

Two references have been added that explain this. The topic of governance and its impact on communal land development is discussed in the book "Untitled: Securing Land Tenure in Urban and Rural South Africa," specifically in the Introduction chapter written by Hornby, Royston, Kingwill, and Cousins as well as in the book "Rights to land: A guide to tenure upgrading and restitution in South Africa" by W., Delius, P., and Hay, M.

These books are cited in section 2.4

The authors argue that appropriate governance structures and procedures are essential for successful collaborative decision-making, accountability, and equitable benefit and cost distribution in communal land development. They also emphasize the negative consequences of inadequate governance, including conflict, resource depletion, and unequal distribution of benefits. The authors acknowledge the complexity of the process, which involves intricate interactions among economic, social, and environmental factors.

Reviewer 2 Report

I really like your scientific research. Its research purpose is clearly stated, an appropriate method of approach is applied, and the limitations of the study are explained. Also, prospect for further research is provided.

Author Response

Thank you. The results have been rewritten in Section 4.

Reviewer 3 Report

I have attached my comments.

Author Response

  1. Introduction

Under Introduction line 29-31, the authors talk about the United Nations without talking about the position of the South African government on land issues. What is the land tenure system in South Africa like?

The Introduction has been updated to, among others, provide an overview of land reform in South Africa. Refer to lines 35 to 48.

The authors also talk about “autocratic land governance system” 52. What exactly does this mean? What are the features’ of autocratic land governance system?

While some members of the community perceived the church as conservative and ‘autocratic’, we agree it would be inaccurate to describe it as such. The church operates under a hierarchical structure, which has now been clarified in the text. The word ‘autocratic’ has therefore been completely omitted from the manuscript.

  1. Literature Review

From 107 through184, the authors brings on board several theories. However, some of these theories have not been linked with the study in question. For example there are no mention of other successful case studies in Africa where some of these theories were applied or have worked. So the big question here is how are these different theories useful to the study?

The literature review has been revised to, among others, highlight different studies that utilized the theories adopted in the study. The literature review also includes a subsection on the synthesis to make a connection between the underlying concepts and theories.

What the authors don’t tell us is that a larger middle class actually leads to land grabbing than “support[ing] the rule of law and accountability”138-139 as the authors put it.

It is acknowledged that the statement oversimplified the connection between the middle class, the rule of law, and accountability and thus is reductionist. As a result, Section 2.2 has been rewritten.

Under 2.3(148) the authors should have discussed more on “access to land in south Africa” and “how it is multilayered and poorly understood”. Perhaps this would make the reader to understand the “autocratic land governance system” which kept coming up throughout the paper. Until this is done, the paper is not easy to follow and understand.

In Section 1 (Introduction), the issue of access to land and the issues of land reform in South Africa has been added to give context to access to land. Additionally, the introduction highlights the tension between the dominant private property paradigm and informal land governance systems that provide tenure security but lack official recognition. In Section 2.3, an addition has been added that explains that the critical realist framework aims to explain why and how things are happening, including potential risks and benefits. In this study, power dynamics and conflicts may impact community transition to a communal land governance system - the misunderstanding regarding autocratic land governance has been addressed.

What is the meaning of “theory development” in this paper? 157. The flow of the paper is also want. For example the line/theory from 158-159 comes from nowhere. This makes it hard to follow the logical flow and argument of the paper/theories.

The passage describes the functioning of causal mechanisms in explaining observable events and highlights their inability to be directly observed. It further proposes that empirical research and the development of theories can aid in comprehending these causal mechanisms.

The mention of Bhaskar's critical realism provides a philosophical basis for this argument and contrasts it with the approach of positivism, which seeks to identify general laws. Therefore, the reference to Bhaskar is necessary in this context.

In conclusion, literature review should be redone. What have others written about their study area/research? What were the key issues in those papers? And what were the gaps, and how does their research try to address those gaps?

The literature review has been revised to, among others, highlight different studies that utilized the theories adopted in the study. The literature review also includes a subsection on the synthesis to connect the underlying concepts and theories.

  1. Methods 2

I feel that the information provided from line 193-199 should have come earlier under introduction. That gives the reader the “background to the study” and makes the subsequent reading easy to follow.

The study area has now been mentioned in the introduction (lines 71-74), and cross-referencing is made to Chapter Three, where the study area is presented in detail.

However, I also find contradiction in the second paragraph (201) with the first paragraph. How can hybrid land governance system be dominated by a single organization?

The reference to a hybrid/autocratic land governance system has been omitted, and reference is now made to hierarchical land governance. 

In next sections on findings and discussions, the authors state that the church (state) is very autocratic and does not involve other stakeholders in the planning.

The word ‘autocratic’ has been omitted from the manuscript.

Under 3.2 the authors talk about “institutions and actors” 209. Aren’t these the same? I mean cant institutions be actors and vice versa?

While institutions and actors may overlap somewhat, they are not precisely the same thing. However, to avoid confusion, the word ‘actors’ has been replaced with ‘stakeholders'.

Sixty nine households were interviewed. What then was the sample size, did any member if the household take part in this research? Then at what point was saturation point used to determine the sample (228)? In this case are we talking about the sixty nine households as the saturation point or what? More explanation is needed here.

Simple random sampling was used to select a sample of households from the mission station before the survey was conducted. Each household was assigned a unique identifier using Geographic Information System (GIS) software. Assuming a confidence level of 95%, a margin of error of 5%, and a population size of 500 households, a sample size of approximately 217 households was envisaged. A discussion related to sampling is included in Section 3.2.1.1.

 It is now acknowledged in Section 3.2.1.1 that because of a low response rate (largely due to Covid-19-related reasons), the study is prone is a non-response error. As the community showed high similarity, Section 3.2.1.1 also argues that the saturation point might have been reached during the survey.

Again, what was the composition of the sample in terms of gender and other socio demographic factors? These are missing.

The study noted that the participants were 21 years or older, with 55% below 65 years old and 45% retired and over 65. There was a high degree of similarity or sameness in terms of characteristics in terms of socio-economic status - there was no recording of the gender of the heads of household (Section 3.2.1.1)

The whole section under the analytic methods (256) is not very clear to understand i.e. sentiment analysis (257), the critical realist framework (283), and philosophical analysis

See the revised findings in Section 4

4 instead of 3. Findings

4.1. Key considerations for Developing a Conceptual Framework for Communal Land Governance and Development – has been moved to the literature review and consolidated into the new Section 2.4.

4.2. Conceptual Framework for Communal Land Governance and Development – has been moved to the literature review and consolidated into the new Section 2.4.

4.1

Under this sub section, the authors talk of “the community” in passing. What is a community? Are they homogeneous to have the same views? Were the youth, or women, or working class etc. saying the same thing with one voice?

The paper adopts a simplified definition of community: a group of people living in the same place. This has been clarified in the introduction (lines 40-50) after the first mention of the word community.

Though not homogenous, the community of Goedvawacht has a high degree of similarity or sameness in terms of characteristics, and this has been clarified in section 3.2.1.1

What is interesting here, yet it is a contradiction later on in the paper is from line 313 to 314. “ The community had a positive attitude towards the existing lifestyle and governance system”. This governance system is what the authors call “autocratic system” later on in their findings. Which is which now?

This has been rephrased, and the word ‘autocratic’ omitted.

Isnt central government part of the institutions which govern the community in one way or the other? 344.

This has been rephrased - The Moravian Church and the community association are two of the predominant institutions in the community, the Moravian Church with its governing council and the community association comprising various groups in the community.

How is the church autocratic over its property? Without explaining why and how the church is autocratic, I feel you are biased towards the church. But again, world over, the church has clear guidelines on how its properties are managed.

This has been attended to in the findings, and the word autocratic omitted from the manuscript.

The authors are not telling us about the findings here, rather they are repeating what they have stated already in this section. See 366- 371

Greater clarity has been added to the findings: Section 4.2

How do the issues raised from 372 to 414 relate to land governance? What is the position of the church to these allegations? Did the authors take time to interview the church leadership or not? If yes, what were their responses, because the voice of the church is missing from the whole work?

The church participated in the qualitative interviews, which has now been clarified in Section 3.2.1.3. The findings have also been updated to reflect the side of the church (see Table 9)

5 instead of 4. Discussion

The introduction of these conceptual frame works are misplaced here. See the statement made from 457 to 476. But in concrete terms, what is the relevance of this framework yet you have stated its inadequacy 472- 476

The authors acknowledge the framework’s inadequacies and limitations but propose its use because it is still helpful for understanding and evaluating communal land governance and development. It is argued that no framework can be perfect.

What are the roles of the other stakeholders when it comes to successful communal land governance and development? 462-464

Additional information has been added in section 2.4

The discussions being presented on page 14(490-543) would only work if the land belongs to the community. In this case the land belongs to the church. How do you want the community to manage land which does not belong to them?

This issue has been clarified in this section - The church's land reform obligations drive this transition.

See section 1 (introduction) Land reform typically involves legal and policy changes recognizing and securing communities' land rights. These changes may involve transferring land ownership from private landowners to the community through different legal mechanisms.

Secondly, the authors keep proposing the use conceptual frameworks which they critique at the same time. Which conceptual framework should be adopted then? 531-544

The authors recognize that the framework may not universally apply to different settings and communities and suggest that it be tailored to specific contexts through ongoing research, consultation, and revision. Refer to Section 2.4

The authors are also mixing up things. Can we say that the state is the church in this study (572)? Does the state, as they claim, lack legitimacy. What is at stake now? What powers/or how are the church leaders misusing their power? Power to do what, (mis)manage which is for the church?

Yes, this section was confusing – it has been rewritten in Section 5 (Discussion)

Conclusions

In your conclusion, you make reference to “autocratic hybrid land governance” 367 and 644. The only problem is that you didn’t dedicate a section to unpack what these authoritarian practices are. You also failed to talk about property rights and ownership in South Africa. That leaves this paper one sided and sounding very biased.

The word ‘autocratic’ has been omitted from the manuscript. The Introduction has been updated to, among others, include an overview of land reform in South Africa. Refer to lines 35 to 48.

Reviewer 4 Report

The paper presents an interesting way of analyzing the land governance system in regard to the community’s ability and perception.

My overall remark is that the text needs a clearer form (clarification). In particular, the methods section is to overall, it will be helpful for Readers, to present clearly the procedure of conducting analysis, frameworks adopted, etc.. For example, the Authors use the Fukuyama development framework, but it appears only in the discussion.  

Other remarks:

line 28: statement is not precise enough: Through Sustainable Development Goal indicator five - which one exactly? It would be better to introduce it

line 164/165 and line 168/169 are almost the same/repeated sentences

line 560: The development framework outlines ....through six dimensions, and introduced are (line 567: Ultimately, the fifth dimension reinforces the notion that 567 the rule of law checks the state's power

In Figure 1, on the map with provinces is not indicated the analyzed province

in the Findings section, the research sample is not presented, and there is a lack of a short statistical description of respondents/ groups

in my opinion, figures are not readable enough, especially fig.6 as the main result of the analysis should be prepared graphically better (bigger..)

also following terms need a clear definition: hybrid land governance, autocratic land governance, sustainable land governance, collective and communal land governance 

What is the synthetic answer to the research question: what are the roles, expectations and management strategies of the actors participating in land governance? (line 55/56) 

And what are the economic, social, and political factors that shape land governance? (line 59/60)

Good luck with the resubmission!

Author Response

My overall remark is that the text needs a clearer form (clarification). In particular, the methods section is to overall, it will be helpful for Readers, to present clearly the procedure of conducting analysis, frameworks adopted, etc.. For example, the Authors use the Fukuyama development framework, but it appears only in the discussion.  

The method section has been revised; reference to Fukuyama is now included in the literature review section.

Other remarks:

line 28: statement is not precise enough: Through Sustainable Development Goal indicator five - which one exactly? It would be better to introduce it

This has now been clarified in lines 31 to 35.

line 164/165 and line 168/169 are almost the same/repeated sentences

The duplication has been corrected.

line 560: The development framework outlines .... through six dimensions, and introduced are (line 567: Ultimately, the fifth dimension reinforces the notion that 567 the rule of law checks the state's power

There are five dimensions of development – the correction has been made

In Figure 1, on the map with provinces is not indicated the analyzed province

Figure 1 has been updated to mark the Western Cape province with a rectangle.

in the Findings section, the research sample is not presented, and there is a lack of a short statistical description of respondents/ groups

Findings have been improved in Section 4 (Findings)

in my opinion, figures are not readable enough, especially fig.6 as the main result of the analysis should be prepared graphically better (bigger..)

This has been corrected in Section 4, Figure 4.

also following terms need a clear definition: hybrid land governance, autocratic land governance, sustainable land governance, collective and communal land governance 

The expression "hybrid land governance" is unclear and has been reworded, and its meaning clarified in section 2.1. While some community members perceive the church as conservative and autocratic, it is inaccurate to describe it as such. The church operates under a hierarchical structure, as clarified in the text. Collective and communal land governance was used interchangeably which was incorrect. There are similarities between the two concepts however they are not interchangeable because they refer to different types of ownership and decision-making structures. Collective has been replaced by communal land governance.

What is the synthetic answer to the research question: what are the roles, expectations and management strategies of the actors participating in land governance? (line 55/56) 

The roles, expectations and management strategies of the stakeholders participating in land governance have now been summarized in Section 4.4 Summary of the findings.

And what are the economic, social, and political factors that shape land governance? (line 59/60)

Economic, social, and political factors that shape land governance have been addressed in Section 4.3 and Section 4.4 Summary of the findings.

Round 2

Reviewer 1 Report

The authors have made the changes in proper way